# Pre-Partum Supplementation with Polyunsaturated Fatty Acids on Colostrum Characteristics and Lamb Immunity and Behavior after a Mild Post-Weaning Aversive Handling Period

**DOI:** 10.3390/ani12141780

**Published:** 2022-07-11

**Authors:** Xavier Averós, Itsasne Granado-Tajada, Josune Arranz, Ignacia Beltrán de Heredia, Laura González, Roberto Ruiz, Aser García-Rodríguez, Raquel Atxaerandio

**Affiliations:** 1Animal Production Department, NEIKER-Basque Institute for Agricultural Research and Development, BRTA-Basque Research and Technology Alliance, Campus Agroalimentario de Arkaute s/n 01192, 01192 Arkaute, Spain; igranado@neiker.eus (I.G.-T.); jarranz@neiker.eus (J.A.); ibeltran@neiker.eus (I.B.d.H.); rruiz@neiker.eus (R.R.); aserg@neiker.eus (A.G.-R.); 2Animal Production Department, Centro de Investigaciones Agrarias Mabegondo, Carretera de Betanzos a Mesón do Vento Km. 7,5, Abegondo, 15318 A Coruña, Spain; laura.gonzalez.gonzalez@xunta.gal

**Keywords:** dairy sheep, pre-natal supplementation, linseed, colostrum, immunoglobulins, cytokines, post-natal stress

## Abstract

**Simple Summary:**

Livestock farming faces the necessity of reducing the use of medicines, particularly antimicrobials, without compromising animal welfare as one main challenge. A strategy to improve animal health, specifically in breeding stock, relies on improved nutrition of the mothers during late pregnancy and on management factors during early life. These could affect mother and offspring physiological functions (calostrogenesis, fetal development or maturation of lamb’s immune system) and potentially modulate their stress-response capabilities. We studied the effect of pre-partum omega-3 (n-3) polyunsaturated fatty acids supplementation on ewe colostrum composition and immunological quality and whether changes resulted in any advantage on lambs’ passive immunization and on their physiological, behavioral, and inflammatory responses when subjected to a post-weaning stressor to confer stress resilience. Gestational supplementation with nutrients rich in essential fatty acids was effective to enhance colostrum quality related to biologically active molecules, such as conjugated linoleic acid, which plays a role in the regulation of the immune response. Stress resilience was studied by assessing lamb’s behavioral response after exposing lambs to a combination of stressors and by measuring plasma cortisol concentrations and immunological responses to specific circulating cytokines. This work highlights the complex relationship between late pregnancy nutrition in the mothers and immune system development in the offspring and explores the potential for programming nutrition interventions to develop more resilient livestock and to tackle the need for a reduction in the use of antibiotics.

**Abstract:**

We studied the effect of pre-natal supplementation with n-3 α-linolenic acid (ALA) combined with a tannin-rich forage on colostrum composition and immunological quality and whether these changes had advantageous effects on lambs’ survival and stress reaction to a post-weaning stressor. Forty-eight Latxa ewes were fed during the last five weeks of pregnancy with two experimental diets: a control diet based on a neutral concentrate and forage (tall fescue hay; CO-FES), and a supplemented diet based on polyunsaturated (PUFA)-rich concentrate and tanniferous forage (sainfoin; ALA-SAIN). After parturition, twenty ewes had their blood and colostrum sampled, and their lambs were monitored until post-weaning. Lambs were afterwards subjected to (i) an aversive handling period (AHP) followed by a behavioral assessment and (ii) inflammatory and lymphocyte proliferation challenge. Feeding ALA-SAIN resulted in changes in colostrum fatty acid composition, specifically higher α-linoleic acid (*p* < 0.001), conjugate linoleic acid (*p* = 0.005), vaccenic acid (*p* = 0.006) and long-chain n-3 PUFA (*p* = 0.004). Pre-partum nutrition did not affect lamb immunoglobulin (Ig) G apparent efficacy absorption, but circulating IgG tended to be higher (*p* = 0.054) in ALA-SAIN lambs. ALA-SAIN lambs interacted more frequently with other lambs (*p* = 0.002), whereas ALA-SAIN females spent more time closer to other lambs (*p* < 0.001). Plasma cortisol was higher (*p* = 0.047) and plasma interleukin (IL)-2 lower (*p* = 0.003) in CO-FES lambs. This research highlights the importance of prenatal nutrition on the immune system stimulation and lambs’ behavior as a strategy to improve lambs’ health and welfare during early life.

## 1. Introduction

Production systems are moving towards reducing the use of medicine, particularly antimicrobials, and this requires raising healthier and more resilient animals [1]. Within this context, strategies to enhance lamb resilience in sheep production are essential, with maternal nutrition attracting special interest due to its programming effects on offspring that extend into post-natal life [2,3]. Pre-natal diet supplementation with omega-3 polyunsaturated fatty acids (n-3 PUFA) has been proven to benefit ewes’ immunity due to their anti-inflammatory, antioxidant, immunomodulatory and antimicrobial properties [4,5]. Beyond the mother, the benefits of pre-natal n-3 PUFA supplementation may extend to the lamb, either through direct effects on fetuses during gestation and/or by the consequences of supplementation on the colostrum.

Colostrum is the first nutrient source for lambs and, given that gestating ewes canot transfer immunoglobulins (Ig) to the fetus through the placenta [6], colostrum also plays a protective role, as lambs’ passive immunization exclusively depends on intestinal Ig absorption after birth [7], conditioning their short- and long-term health [8,9]). Therefore, the modulation of colostrum properties via pre-natal maternal nutrition is a promising approach to improve lambs’ health. Pre-partum n-3 PUFA supplementation positively modulates colostrum composition and fatty acid profile [10], while supplementation with conjugated linoleic acid (CLA) has additionally been found to enhance neonates’ immune system [8], likely through increased Ig absorption, as observed in calves [11]. Nevertheless, experiments testing the benefits of n-3 PUFA supplementation on colostrum production are not conclusive [12,13], and thus their eventual implications on lamb immune response remains uncertain [14].

Besides health, pre-natal n-3 PUFA supplementation may also benefit lambs’ brain development, early behavior and survival rate [15], also both via pre- and/or post-natal influence. In the latter case, this will most likely occur due to the resulting modifications in colostrum fatty acid profile [16]. A link exists between early life intestinal programming mechanisms and stress responsiveness, so that intestine upward communication would modulate brain activation, as well as the different physiological and behavioral responses to stress [17]. The early intestinal modulation of brain activity depends on, among others, intestine nutrient availability. In consequence, pre-natal development conditions, colostrum composition and feeding management would not only play a key role on lambs’ immunity but also on how they cope with different stressors [18]. Given the interactions between the intestine, immune system functionality and stress response [19,20,21], it may be hypothesized that modifications in the colostrum fatty acid profile, achieved through late pregnancy supplementation to the mothers, will also benefit lambs’ stress-coping abilities and welfare. Given that stress may interfere with the normal immune response [22], it can be expected that promoting enhanced coping abilities and welfare will reduce the negative consequences that stress has on the normal immune system functioning, which will further benefit animal health. Therefore, enhanced resilience would be achieved through the improvement of animal health and welfare, aligning with the One Health concept [23].

The aims of this study were to determine whether pre-partum n-3 PUFA supplementation results in changes in ewe colostrum composition and immunological quality and to determine whether these changes result in advantages on lambs’ passive immunization and on their physiological, behavioral and inflammatory responses when subjected to a post-weaning stressor.

## 2. Materials and Methods

### 2.1. Animals, Diets, and Management Pre-Partum

The experimental flock of Neiker (Arkaute, Araba-Álava, Spain; latitude 42°51′5″ N, Longitude 2°38′6″ W) was used in the study, conducted from December 2017 to April 2018. All procedures followed Directive 2010/63/EU (European Union, 2010) on the protection of animals used for experimental and other scientific purposes and were also approved by the Neiker Ethics Committee (NEIKER-OEBA-2017-007). A total of 48 multiparous pregnant ewes from the Latxa breed, artificially inseminated in August 2017, were divided into 4 groups of 12 ewes and balanced according to their breeding value, prior lactation milk yield, body condition score (BCS) recorded one month before the expected date of birth and parity number. Groups were randomly assigned to 1 of 2 pre-natal diets (2 replicates/diet) and then to 1 of 4 indoor pens (2.1 m^2^/ewe); they were provided straw bedding and free access to water through a nipple drinker, and the ewes remained there from week 6 prior to birth until the end of the lambing period. During the pre-partum period, groups were fed 3 times/day with a control (CO-FES) or a supplemented (ALA-SAIN) diet. The CO-FES diet consisted of a control concentrate (CO; 450 g dry matter (DM)) containing hydrogenated fat (from prilled palm) plus tall fescue hay (*Festuca pratensis*; FES) ad libitum; the ALA-SAIN diet consisted of a polyunsaturated fatty acid rich concentrate (450 g DM) containing linseed rich in α-linolenic acid (C18:3n-3; ALA; Tradilin^®^, Valorex, Combourtillé, France) with a 20% inclusion rate, cold-pressed cake rapeseed (CPCR) with a 5% inclusion rate and sainfoin hay (*Onobrychis viciifolia*; SAIN) ad libitum. Diets were complemented with 750 g DM corn silage/ewe and day as an energy extra source. Concentrates were formulated to provide equal amounts of fat and energy. Concentrate protein was balanced according to the amount provided by the forage (Appendix A). Concentrate was offered in the morning in individual feeders, and refusals were recorded per ewe. Corn silage was offered in the morning and forage in the afternoon, with daily amounts adjusted to allow 0% and 10% refusals, respectively, on a feeding tape allowing simultaneous access (0.5 m/ewe).

### 2.2. Animals, Management and Measurements from Birth until Weaning

During the lambing period, only lambings occurring between 6:00 a.m. and 4:00 p.m. could be completely monitored. Therefore, 22 out of 47 births (1 non-pregnant ewe) could be completely controlled. Lambs from the first and last births were discarded to balance lamb sex within groups, and lambs from the remaining 20 births were further monitored. Immediately after birth, lambs were individually identified, weighed and separated from their mothers. In case of multiple (twins and triplets) births, the first lamb born was retained for the experiment.

#### 2.2.1. Ewes

Two colostrum samples were collected from each ewe, at birth and 24 h post-partum, by manual milking until the complete depletion of the udder. Colostrum total volume (mL), density (g/mL; Brine density meter) and pH (pH meter, Hanna Instruments, Villafranca Padovana, Italy) were determined within 2 h post-partum at room temperature (21 ± 3 °C). Colostrum somatic cell counts (SCC) were also determined on fresh samples at birth and 24 h post-partum with a portable cell counter (DeLaval, Madrid, Spain) following Gonzalo et al. (2006) [24]. Briefly, 1 mL colostrum was diluted 1:1 in phosphate-buffered saline (PBS) + Triton X-100 (0.2%) + propidium iodine (0.025 mg/mL; Sigma-Aldrich, Madrid, Spain). A soak time of 1 min was employed before cell counting.

A 100 mL birth colostrum aliquot was collected on azidiol (concentration 0.33%; [25]) to determine chemical composition using normalized methods. Variables were % protein (Kjeldahl method), % fat (Röse-Gottlieb method), % dry extract (ISO 6731:2010/IDF 21:2010) method), % ash (OM 1694 method) and % lactose (calculation method; Instituto Lactológico Lekunberri, Lekunberri, Spain). Two additional 10 mL birth colostrum aliquots were frozen (−20 ± 5 °C) for further measurements.

One aliquot was stored with azidiol (0.33%) for fatty acid profile determination, then thawed for colostrum fat extraction and fatty acid derivatization to fatty acid methyl ester (FAME) (following basic methylation using methanol KOH) using normalized methods (ISO 1885:2002/IDF 184:2002). The FAME was analyzed by gas chromatography (GC) in a Varian 3900 GC (Varian, Instruments, Walnut Creek, CA, USA) equipped with a flame ionization detector (FID), a BPX70 capillary column (120 m × 0.25 mm i.d., 0.25 μm film thickness; SGE, Trajan Scientific, Ringwood, Australia) and a Varian 8410 autosampler.

The injector and detector were kept at a constant temperature of 250 °C. The column oven temperature program for FAME separation was adapted from the methodology described by Kramer et al. (2002) [26] (45 °C for 4 min, then heated by 13 °C/min to 175 °C, kept there for 27 min, heated by 4 °C/min to 215 °C and kept there for 35 min, with a total run time of 85.62 min). The carrier gas used was helium at a constant flow of 1.3 mL/min.

The split ratio was 20:1, and 1 µL of the sample was injected. With this oven program, some *trans(t)*-18:1 (*t*-18:1) isomers and CLA isomers were not well resolved. The further separation of *trans*-18:1 isomers would have required separation with silver-ion thin-layer chromatography or solid-phase extraction, followed by isothermal GC using a relatively low temperature [27].

In the same way, a complete CLA isomer profile would require an Ag+-HPLC analysis according to Cruz-Hernandez et al. [28]. These specific analyses were not carried out, as with this column and in a simple run we were able to identify/quantify the main ones in the milk fat, allowing the separation and identification of the *t*6-*t*9C18:1, *t*10-18:1, *t*11-18:1 and *t*12-18:1 isomers, as well as the sum of *cis(c)*-9, *trans*-11-octadecadienoate (*c*9,*t*11CLA or rumenic acid, RA) + *t*7,*c*9 CLA + *t*8*c*10 CLA (assuming *c*9,*t*11CLA is over 95% of these three isomers).

The FAME peaks were identified using the FAME mix C4-C24 and methyl *trans*-11-vaccenate standard acquired, both from Supelco Inc. (Bellefonte, PA, USA), and methyl *c*9, *t*11-CLA acquired from Matreya LLC (State College, PA, USA). The GLC-603 and GLC-463 standards from Nu-Check Prep, Inc. (Elysian, MN, USA) were used to identify the FAME not contained in the FAME mix. Branched-chain FAME were identified using a GC reference standard BC-Mix1 purchased from Applied Science (State College, PA, USA). Additional FAME not present in the standards were identified based on their elution orders and GC/MS identification [28].

The quantification of FAME was based on peak area, with calculations using 3 internal standards: methyl nonanoate (C9:0) and methyl *cis*-10-heptadecenoate (*c*10-C17:1) (both from Sigma-Aldrich, Spain) and methyl *trans*-10, *cis*-12-octadecadienoate (*t*10,*c*12-CLA; Matreya LLC, State College, PA, USA) according to the wide range of concentrations within the FA profile. Moreover, the methyl nonadecanoate (C19:0) (acquired from Sigma-Aldrich Co., Madrid, Spain) was used as a surrogate.

The other 10 mL frozen aliquot and the 24 h post-partum 10 mL frozen aliquot were thawed at room temperature (21 ± 3 °C) and used to determine colostrum total solids concentration (°BRIX; optical-digital refractometry; Hanna Instruments, Villafranca Padovana, Italy) and IgG concentration (radial immunodiffusion (RID) on solid plaques; IDBiotech SARL, Issoire, France). Colostrum leptin and lysozyme concentrations were only determined on the colostrum aliquot obtained right after birth using commercial ELISA kits, following the manufacturer’s instructions (MyBioSource, San Diego, CA, USA). Briefly, colostrum samples were centrifuged for 20 min at 1000× *g* and 4 °C, and the resulting serum under the fat layer was analyzed.

Two blood samples were collected on tubes containing EDTA from ewes right after birth. One sample was used for hematological analyses: erythrocytes (10^6^ cells/μL) and leukocytes (10^3^ cells/μL), hematocrit (%), hemoglobin (mmol/L), mean corpuscular volume (MCV; fl), leukocyte differential cell counts (%) and neutrophil lymphocytes ratio (N:L) using an electronic counter (Hemavet 950, Drew, MS, USA). The other sample was centrifuged (2000× *g*, 10 min), and the resulting plasma was stored at −80 °C until analysis of concentrations of cortisol and interleukin IL-2 (competitive -ng/mL, and sandwich immunoassay -pg/mL, respectively; Cusabio, TX, USA), and leptin and lysozyme (-ng/mL and µg/mL sandwich immunoassay, respectively; MyBioSource, San Diego, CA, USA), according to manufacturer’s instructions.

#### 2.2.2. Lambs

Experimental lambs (10 males and 10 females) were moved to 1 of 2 identical, adjacent post-partum pens where they remained until the end of the study. Pens were 12 m^2^, had straw bedding, drinking water ad libitum through 1 nipple drinker and walls lined with black plastic to avoid visual contact outside the pen. Within each pen, lambs were balanced according to pre-natal treatments, with sex balanced within treatment. First, lambs were bottle-fed a colostrum volume corresponding to 10% of birth body weight (BW) [29] using a maximum of 3 periods within the first 18 h after-birth. After finishing colostrum, lambs were allowed ad libitum access to a milk replacer (Agno-Chevro 63, Celtilait, Ploudaniel, France) through 2 nipple drinkers/pen.

Lambs were weighed at lambing, 7 d, 14 d, 21 d, 28 d and 35 d of age. Blood samples were obtained at 24 h, 48 h, 7 d and 21 d of age on evacuated tubes (Vacutainer^®^Serum, BD). Samples were allowed to clot at room temperature and centrifuged at 2500× *g*, and the serum stored at −20 ± 5 °C until analysis. Lamb serum IgG concentration at 24 h, 48 h, 7 d and 21 d post-lambing was determined as previously described. IgG apparent efficiency of absorption (AEA) at 24 h and 48 h post-partum was determined following Quigley and Drewry (1998) [30]. At 35 d after the last lambing (BW = 17.0 ± 0.4 kg), all lambs were weaned and moved to solid feeding.

### 2.3. Animals and Aversive Handling Period (AHP) Post-Weaning

After a 5 d adaptation period, lambs were subjected to an aversive handling period for 12 d. During the AHP, lambs were daily subjected to 2 immobilizations (morning and afternoon; 30 min each) based on Destrez et al. (2013a) [31]. To monitor lamb weight during this period, lambs were weighed at 0 d, 6 d and 12 d after starting the AHP (40 d, 46 d and 52 d of age).

### 2.4. Behavior Tests Post-AHP

At 53 d of age, lamb behavior was assessed as described in Averós et al. (2015) [32]. Behavioral tests were carried out in groups of 4, with 2 lambs/pen pseudo-randomly selected to keep a balance between treatments and sexes within each group. Each group was moved from the home pen to a pre-test area where the lambs remained for 5 min to standardize initial conditions. One lamb was then randomly selected from the group and moved to a contiguous arena (area 2.5 m × 3 m) with straw bedding and walls lined with black plastic to avoid initial visual contact with other lambs. The lamb was placed at the starting point and subjected to a 5 min social isolation test, during which lamb behavior was collected by direct observation using instantaneous scan sampling. For each scan, lamb position (XY coordinates) and behavior were collected using the Chickitizer software [33]. Behaviors were: passive stand (standing immobile on 4 feet), move (changing position within the pen, either walking or running), explore (nose interaction with the floor or walls) and escape attempt (jumping towards the pen walls, actively attempting to escape). Total number of excretions (n) and vocalizations (n) were collected by continuous sampling. At the end of the test, the lamb was returned to the starting point and allowed to gain visual access to the other 3 lambs. Then it was subjected to a 5 min social motivation test. Collected behaviors were: passive stand, move, escape attempt and interaction with other lambs (physically interacting with the other lambs through the fence), collected using instantaneous scan sampling. Latency to get to the fence (s), time spent close to the other lambs (s), vocalizations (n) and total excretions (n) were collected using continuous sampling. After this, the lamb was returned to the pre-test area. The 4 lambs from each group were sequentially tested as described and then moved back to their respective home pen. After this, a new group was identically tested. For each test, lamb movement trajectory and space use were determined using XY coordinates. An estimate of the error associated with the data collection was obtained as described in Averós et al. (2014, 2015) [32,34], and lambs’ positions were corrected accordingly. From the corrected positions, total moved distance (cm) and angular dispersion, as a measure of the trajectory tortuosity, were estimated [34,35,36,37]. Behavior frequencies (% of total sampled scans) were also calculated. For social motivation tests, time spent close to other lambs, relative to total test time (%), was additionally calculated.

### 2.5. Inflammatory and Lymphocyte Proliferation Challenge (IC) Post-AHP

At the end of the tests, the lambs were subjected to an inflammatory and lymphocyte proliferation challenge (IC). Each lamb was subjected to the intradermal injection of 1 mL phytohemagglutinin (PHA; Sigma-Aldrich, Madrid, Spain), with a concentration of 1 mg/mL in a sterile saline solution [38]), administered in the center of a pre-shaved, 2 cm diameter skin circle on the right scapular area. The skinfold thickness (mm) of the injection area was determined pre-injection, 24 h, 48 h and 72 h post-injection with a calibrated calliper (3 replicates/measurement). Lymphoproliferation in the injection area at 24 h, 48 h and 72 h post-injection was calculated as the difference between 24 h, 48 h and 72 h thickness and pre-injection thickness. Changes in surface skin temperature (°C; laser surface temperature meter; PCE Instruments, Tobarra, Spain) at the point of injection were recorded during the same periods. Lambs were blood-sampled after behavioral tests, 24 h and 48 h post-IC (53 d, 54 d and 55 d of age, respectively). Cortisol concentration was obtained from all plasma samples as previously described. Changes in main cytokines related to T-cell proliferation and inflammatory function were also assessed. Therefore, IL-2, IL-10 (competitive) and IL-1β (sandwich immunoassay) concentrations (pg/mL) were also determined on 24 h and 48 h post-IC samples using commercial ELISA kits (Cusabio, TX, USA).

Mortality and health issues with lambs were recorded throughout the duration of the study.

### 2.6. Statistical Analysis

Effects of independent variables on each dependent variable were analyzed using generalized linear mixed models with the GLIMMIX procedure in SAS version 9.4 (SAS Institute, Cary, NC, USA). Independent variables were adjusted to a Gaussian distribution unless otherwise stated, and their residuals assessed for normality and variance homoscedasticity. In case residuals did not distribute normally, independent variables were assumed to follow a lognormal distribution. Ewe hematocrit (%), behavior frequencies (%) and time spent in each pen region with respect to total social motivation test time (%) were adjusted to a binomial distribution, whereas excretions and vocalizations (n) were adjusted to a Poisson distribution. In case of repeated measures analysis, the animal was included as the repeated measures unit, and a first-order autoregressive covariance structure was assumed to account for any linear dependence of data over time. Least-square means were computed in case of statistically significant effects (*p* < 0.05), with *p*-values adjusted for multiple comparisons by Tukey range tests. For significant interactions, tests of simple effects [39] were performed to detect differences between the levels of one independent variable within each level of the other independent variable (*p* < 0.05). Additional details about independent variables included in the statistical model for each dependent variable are provided in Appendix A.

## 3. Results

### 3.1. Pre-Partum Concentrate Intake

Pre-partum individual concentrate ingestion was 98.0% in CO-FES ewes (82.2–100%) and 98.9% in ALA-SAIN ewes (95.2–100%).

### 3.2. Ewe Hematology and Plasmatic Values at Birth

No effect of pre-natal diet on any hematological variable was detected (Table 1). Ewe hematocrit and hemoglobin levels increased as BCS increased (*p* = 0.017 and *p* < 0.001, respectively) and were higher in single births than in multiple births (*p* = 0.008 and *p* = 0.011, respectively). Ewe cortisol concentration at birth was not affected by pre-natal diet but tended to be higher in multiple births than in single births (205.1 ± 15.1 and 163.8 ± 13.4 ng/mL, respectively; *p* = 0.081). No effect of pre-natal diet was detected on plasma IL-2 concentration at birth, which tended to decrease as gestation duration increased (*p* = 0.081). Plasma leptin concentrations at birth were higher in single births than in multiple births (13.12 ± 1.91 and 4.68 ± 2.16 ng/mL, respectively; *p* = 0.021), similar to plasma lysozyme concentrations (30.45 ± 4.17 and 13.80 ± 4.72 µg/mL, respectively; *p* = 0.034).

### 3.3. Birth to Weaning Period

#### 3.3.1. Colostrum Physical Measurements, Chemical and Fatty Acid Composition

No effect of pre-natal diet on colostrum physical measurements and chemical composition was detected (Table 2), except for a trend toward lower fat content in ALA-SAIN ewe colostrum. On the other hand, changes were detected in the colostrum fatty acid composition profile according to pre-natal diet (Table 3), with total *t*-MUFA, total PUFA and long-chain n-3 PUFA (equal to or higher than 18C) concentrations being higher in ALA-SAIN colostrum (*p* < 0.05). Remarkably, α-linolenic acid (*c*9, *c*12, *c*15-18:3, 18:3n-3 or ALA) (*p* < 0.001), conjugated linoleic acid (∑CLA) (*p* < 0.01) and vaccenic acid (*t*11-18:1) (*p* < 0.01) concentrations in ALA-SAIN colostrum were substantially higher than in CO-FES colostrum. Consequently, the C18:1 *trans*-11/*trans*-10 ratio was affected on colostrum from ewes fed ALA-SAIN pre-natal diet (*p* < 0.001). Although pre-natal diets did not alter long-chain n-6 PUFA concentrations, the ratio n-6:n-3 was also altered in ALA-SAIN colostrum (*p* < 0.01). Regarding saturated fatty acids, pre-natal diets slightly changed fatty acid profile, and only lauric acid (C12:0) and pentadecyl acid (C15:0) concentrations were higher in ALA-SAIN colostrum (Appendix A).

#### 3.3.2. Colostrum SCC, Total Solids and IgG Concentration

Colostrum SCC concentration (Table 4) was neither affected by pre-natal supplementation nor changed 24 h post-partum. Colostrum total solids and IgG concentrations decreased 24 h post-partum regarding birth values.

#### 3.3.3. Lamb Serum IgG Concentration and AEA

Lamb serum IgG concentration did not change until 48 h post-lambing, decreasing afterwards (*p* < 0.001). This change tended to be affected by pre-natal diet (*p* = 0.054; Figure 1), so that concentration decrease after 48 h post-lambing was slightly more marked in CO-FES lambs, which tended to have lower plasma IgG concentrations at 21 d of age. Neither of the tested variables affected lamb AEA (mean values of 16.19 ± 3.39 and 15.19 ± 3.31% for CO-FES, and 17.59 ± 3.50 and 16.95 ± 3.42% for ALA-SAIN lambs at 24 and 48 h post lambing; *p* = 0.857).

#### 3.3.4. Lamb BW from Birth until Weaning

Pre-natal diet affected lamb BW at birth (*p* = 0.025) with CO-FES lambs (5.24 ± 0.10 kg) being heavier than ALA-SAIN lambs (4.63 ± 0.08 kg). Males (5.18 ± 0.11 kg) were heavier than females (4.73 ± 0.08 kg; *p* = 0.038). Changes in lamb BW until 35 d of age tended to differ according to pre-natal diet (*p* = 0.055; Figure 2), with ALA-SAIN lambs tending to still be lighter at 7 d of age, although this trend disappeared afterwards.

### 3.4. Post-Weaning Period

#### 3.4.1. Lamb BW and Plasma Cortisol Concentration during Post-Weaning AHP

Males showed a slight trend to grow faster than females (*p* = 0.099; Figure 3). A significant pre-natal diet×sex interaction was detected on lamb BW (*p* = 0.026; Figure 4), with the BW of CO-FES lambs not differing according to sex, while ALA-SAIN males were heavier than females. Lamb plasma cortisol concentration during AHP was not affected by any of the tested effects, but a trend to higher cortisol concentration at the end of the period was detected in ALA-SAIN lambs with respect to their initial values (*p* = 0.080; Appendix A), which was not observed in CO-FES lambs.

#### 3.4.2. Lamb Behavior and Plasma Cortisol Concentrations after Behavior Tests

As shown in Table 5, during social isolation tests ALA-SAIN lambs tended to stand passively less frequently than CO-FES lambs (*p* = 0.058) and to vocalize more (*p* = 0.096). Males attempted to escape more frequently (*p* = 0.002); their movement trajectories were less sinuous (higher angular dispersion; *p* = 0.034), and they tended to vocalize more (*p* = 0.064) than females.

During social motivation tests, a significant pre-natal diet×sex interaction was observed on the % of time spent on the pen region closest to the other lambs, so that no pre-natal diet differences were detected for males, while ALA-SAIN females spent significantly more time closer to other lambs than CO-FES females (*p* < 0.001). The number of total excretions was also affected by this interaction, with male excretions not differing according to pre-natal diet but with ALA-SAIN females excreting less than CO-FES females (*p* = 0.022). The pre-natal diet×sex interaction also tended to affect the latency to reach the fence where the other lambs were waiting, so that ALA-SAIN males tended to get close to the other lambs faster than ALA-SAIN females (*p* = 0.090), while latencies for CO-FES lambs were very similar across sexes. ALA-SAIN lambs interacted more frequently with the other lambs (*p* = 0.002), tended to stand passively less frequently (*p* = 0.058), and tended to describe more sinuous trajectories (smaller angular dispersion; *p* = 0.051) than CO-FES lambs. Males tended to vocalize less than females (*p* = 0.079). Lamb plasma cortisol concentration at the end of tests was not affected by pre-natal diet (*p* = 0.801) or sex (*p* = 0.999).

#### 3.4.3. Inflammatory and Lymphocyte Proliferation Challenge (IC) Post-AHP

Skinfold thickness showed significant changes post-IC (Table 6; *p* < 0.001), with values remaining high until 48 h post-IC but decreasing afterwards. Skin temperature also changed significantly post-IC (*p* = 0.038), being higher at 48 h than at 72 h, and being intermediate at 24 h. Plasma cortisol concentration post-IC (Table 7) was higher in CO-FES than in ALA-SAIN lambs (*p* = 0.047). Cortisol tended to be highest immediately after IC and gradually decreased at 24 and 48 h post-IC (*p* = 0.051). Plasma IL-2 concentration post-IC was lower in CO-FES than in ALA-SAIN lambs (*p* = 0.003). A trend was observed for the interaction between pre-natal diet and sex (*p* = 0.050), with the pre-natal diet effect tending to be more apparent in males (55.8 ± 2.1 and 51.1 ± 0.1 pg/mL for ALA-SAIN and CO-FES, respectively) while differences were minimal in females (52.4 ± 0.6 and 51.7 ± 0.5 pg/mL for ALA-SAIN and CO-FES, respectively). Changes in IL-2 post-IC tended to differ according to sex (*p* = 0.078), so that males tended to have higher values 24 h post-IC (55.1 ± 2.8 pg/mL) with respect to values immediately after IC and 48 h post-IC (52.2 ± 1.1 and 52.2 ± 0.7 pg/mL, respectively), while females tended to have higher values 48 h post-IC (53.1 ± 1.0 pg/mL) than immediately after IC and 24 h post-IC (51.8± 0.6 and 51.3 ± 0.2 pg/mL, respectively). Plasma IL-10 and IL-1β concentrations post-IC were not affected by any of the tested variables and interactions.

Throughout the study period, there were no deaths of lambs or any other significant health problems.

## 4. Discussion

The study of interactions between nutrition, immune system stimulation and health and stress regulation is gaining importance in animal production as an alternative to antimicrobial use [5,40]. Pre-natal nutrition, in terms of ration energy, energy/protein balance and supplementation with molecules with bioactive properties, influences colostrogenesis and colostrum quality, as well as lamb intra-uterine growth, post-lambing vitality and immune system activation, although studies are sometimes contradictory [8]. The supplementation of pre-partum diets with PUFA has gained attention during recent years, especially in cattle [41], as a strategy to improve offspring passive immunity transfer (PIT) and protection against infectious and environmental challenges during sensitive phases such as the birth-to-weaning period, as well as to improve maternal health and welfare. However, the effects of ALA supplementation in pre-partum diets in dairy sheep are still poorly understood.

### 4.1. Ewe Hematology and Plasmatic Values at Lambing

Maternal blood variables right after birth were unaffected by pre-natal diets, agreeing with other studies and confirming that optimal nutritional levels do not alter maternal blood parameters at the end of pregnancy [42,43]. Mean hematocrit and hemoglobin concentrations were within normality, although lower values in ewes with lower BCS (data not shown) indicate the importance of an adequate body condition during pregnancy to minimize anemia risk and other health problems. Lower hematocrit, hemoglobin values and plasma leptin concentrations of multiple-bearing ewes reflect their higher physiological demands and associated metabolic attempts to fulfill higher energy demands through increased appetite. Higher cortisol concentrations right after lambing in multiple-bearing ewes would additionally reflect higher partum stress in multiple births. Lysozyme is a mammary humoral defence component that plays a key role in the natural passive immunity transferred to lambs via colostrum. Thus, lower plasma lysozyme concentration in multiple-bearing ewes is likely reflecting higher maternal efforts to achieve adequate immunization for all lambs via lysozyme transfer from maternal blood to colostrum. The trend for decreased plasma IL-2 concentration with gestation length could reflect pregnancy maintenance physiological mechanisms that include reducing IL-2 expression to avoid fetal rejection [44].

### 4.2. Colostrum Physical Measurements, Chemical and Fatty Acid Composition

Both diets were designed to fulfill ewes’ requirements, a fact that is reflected in the lack of differences in those variables related to colostrum synthesis. The use of different fat and forage sources did not alter colostrum nutritional composition either. Colostrum chemical composition is consistent with data from other dairy sheep breeds [45]. However, the fat percentage of ALA-SAIN colostrum tended to be lower than those of ewes fed with a concentrate containing saturated fat. This would agree with the decrease in colostrum fat concentration found by Capper et al. (2006) [15] after supplementation with fish oil from d 103 of gestation, although this study also reported a decrease in milk yield and protein concentration. The fat inclusion level in both diets was close to optimal (8%), and NDF content was above minimum requirements. Therefore, their effect can be discarded, and it may be hypothesized that dietary PUFA influenced ruminal fermentation patterns and modified substrate availability for fat synthesis in the mammary gland [46,47]. Results are in accordance with those reported by Banchero et al. (2006) [48], who did not find any effect of pre-partum diets on colostrum volume and nutritional composition when these fulfilled the nutritional requirements of ewes. On the other hand, Banchero et al. (2004) [49] increased colostrum volume with a 135% increase in the daily metabolizable energy of the pre-partum diet, although this relationship is affected by the nature of diet supplements and their consequences on colostrogenesis [8]. In this sense, ALA supplementation and inclusion percentage did not negatively affect colostrum production and composition.

Coleman et al. (2018a) [16] described modifications in colostrum fat profile after eicosapentaenoic acid (EPA) and docosahexaenoic acid (DHA) supplementation during a pre-partum period slightly longer than ours (55 vs. 30 ± 1.6 days). In the present study, the previously used strategy of using tannin-rich fodder [50] appeared to be successful, as changes in colostrum fat profile were achieved with a shorter supplementation period. Colostrum from ALA-SAIN ewes was richer in *t*-MUFA and PUFA, particularly n-3 PUFA and CLA. Higher *t*-MUFA concentration was mainly due to the increase in circulating vaccenic acid (*t*11-18:1), an RA precursor in the mammary gland, from the action of the enzyme stearoyl-CoA desaturase (SCD) [51]. Thus, the increase observed in *t*11-18:1 and the consequently highest *t*11/*t*10 ratio found by ALA-SAIN consumption could be considered beneficial trough prevention of the undesirable shift to *t*10-t18:1 accumulation during ruminal biohydrogenation, which is related to milk fat depression (MFD) and impaired animal health [52]. Increases in colostrum CLA concentration, specifically *c*9,*t*11-CLA (RA), would be particularly important, given its antioxidant properties and benefits on lipid metabolism and immune function [53,54,55]. Nevertheless, ALA-SAIN colostrum volume was not higher, as was already reported by Castro et al. (2011) [8].

### 4.3. Colostrum SCC, Total Solids and IgG Concentration

No differences in colostrum SCC due to pre-natal diet were found. In this sense, further work would be necessary to improve the characterization of specific colostrum immune cell populations. Mean colostrum IgG concentration was higher than those reported for other sheep dairy breeds but similar or lower than those of sheep meat breeds [45] The selection of dairy breeds for traits such as milk yield, and an associated dilution effect, would explain why their colostrum quality is on average worse. Colostrum IgG concentration was not affected by pre-natal diet, supporting the independence of IgG transfer from pre-partum nutrition, provided that basic needs are fulfilled [8]. The decrease in colostrum IgG concentration 24 h post-partum was also independent from pre-natal diet, similar to what was described by Castro et al. (2006) [56] in goats whose colostrum was not sampled until 96 h post-partum. This deep decrease in IgG concentration, compared to that reported in other studies, would be due to the total mammary gland depletion to estimate colostrum volume right after lambing and would explain the concomitant total solids concentration decrease observed 24 h post-partum.

### 4.4. Lamb Serum IgG Concentration and AEA

Lamb serum IgG concentrations 24 h post-lambing were low in comparison to those of Alves et al. (2015) [29], despite lambs from both experiments being offered the same amount of colostrum (10% of birth BW). The discrepancy between the studies might be due to breed differences in colostrum IgG concentration and lamb IgG absorption capacity, which are apparently lower in the Latxa dairy breed. The mortality after lambing was in any case low, and no health issues affected lambs during the study, indicating that the importance of lamb blood IgG concentration becomes relative in the absence of health challenges during the first days of life [57]. The dynamics of lamb serum IgG concentration until 21 d of age was normal [58], but, interestingly, a slight modulation of serum IgG concentration was detected according to pre-natal diet, with ALA-SAIN lambs tending to have higher values at 21 d of age and therefore slightly enhanced immune protection until they could produce their own IgG at about 1 month of life [59]. On average, AEA values 24 and 48 h post-lambing were much lower than those reported by Alves et al. (2015) [29], and this can be explained by the aforementioned differences in colostrum and lamb serum IgG concentrations. The colostrum feeding method may also have had an influence, as the use of a nipple bottle is associated with less successful passive immunity transfer compared to suckling from mothers [60].

### 4.5. Lamb BW from Birth until Weaning

As expected, males were on average heavier at birth than females. On the other hand, and conversely to previous studies in which pre-natal PUFA supplementation did not affect lamb BW at birth [15,61,62], CO-FES lambs at birth were heavier than ALA-SAIN lambs. These differences might be due to aspects such as fat source or supplementation rate, an aspect already pointed out by Coleman et al. (2018b) [62]. Nevertheless, differences were relatively unimportant, as they disappeared from 14 d of age onwards.

### 4.6. Lamb BW and Plasma Cortisol Concentration during Post-Weaning AHP

The fact that males tended to grow faster during post-weaning AHP was also expected. Nevertheless, mean BW during post-weaning AHP was smaller for ALA-SAIN females with respect to ALA-SAIN males, which would indicate that pre-natal supplementation would accentuate normal sex differences in BW and growth, suggesting a higher susceptibility of ALA-SAIN females to the AHP. Cortisol values also indicate that ALA-SAIN lambs tended to be more sensitized to aversive handling at the end of the AHP.

### 4.7. Lamb Behavior and Plasma Cortisol Concentrations after Behavior Tests

Stress influences animal cognitive abilities and perception and interaction with the environment [63]. Given their short and long-term consequences on animal coping mechanisms, pre-natal interventions may be a good strategy to enhance capabilities to negotiate the stress imposed by productive challenges [64]. For instance, increasing the pre-partum amount of concentrate offered to ewes resulted in lambs with a shorter latency to first suckle [65]. Changes in the coping potential of animals are mediated by pre-natal brain modifications and influence their behavior [66]. In this sense, Capper et al. (2006) [15] found that supplementing pregnant ewes with fish oil resulted in more active lambs, which concurred with parallel modifications in the neonatal brain fatty acid composition. Pre-natal diet effects on lamb behavior were somewhat limited in the present study but still relevant. ALA-SAIN lambs tended to be more active during social isolation tests, and this can be associated with less fearfulness [32]. ALA-SAIN lambs also tended to vocalize more during social isolation tests. Coulon et al. (2011) [67] found that a pre-natal intervention, in that study using a pre-natal stress model, resulted in lambs that vocalized less during behavioral tests, which was interpreted as a sign of passive fearfulness. Results therefore suggest that pre-natal ALA supplementation, probably through the combined effects of pre-natal brain modifications and post-natal colostrum ingestion, tended to result in less fearful lambs after post-weaning AHP. This would confirm the mid-term benefits of pre-natal PUFA supplementation on lamb coping abilities. During social motivation tests, ALA-SAIN lambs also tended to be more active and interacted more with other lambs. The establishment of visual contact with other lambs was expected to modify the behavior of isolated lambs [68]. Results indicate that, under social isolation, establishing visual contact with other lambs made ALA-SAIN lambs more pro-active in seeking social contact, and thus their social motivation was stronger after post-weaning AHP.

Results also reflect sexual differences in lamb behavior. Males were more reactive to social isolation and tended to vocalize more, thus disagreeing with previous results in newborn lambs [32] and adult sheep [69] and suggesting that differences between studies can be attributed to age and previous experience. Sex modulated lambs’ response to social motivation tests, with ALA-SAIN males showing stronger social motivation than CO-FES males, and with CO-FES females being more fearful. Newborn females had higher social motivation than males in a previous study [32], and so discrepancy with the present results suggests that the nature of pre-natal interventions determines how sex will affect the behavioral responses of lambs. Cortisol values at the end of the tests indicate that, contrary to behavior, the physiological response to tests was independent from sex and from pre-natal diet.

### 4.8. Inflammatory and Lymphocyte Proliferation Challenge (IC) Post-AHP

Changes in skin-fold thickness and skin temperature indicate a local, cell-mediated immune response of lambs after PHA injection that was not affected by pre-natal diet or sex. Plasma cortisol concentrations tended to decrease until 48 h post-injection, and plasma IL-1β immediately after injection was numerically higher than values at 24 and 48 h post-IC. Put together, both facts appear to indicate a very slight HPA axis activation right after injection. Otherwise, plasma IL-1β levels remained unaffected by pre-natal diet, but plasma cortisol values were on average higher in CO-FES lambs suggesting higher basal stress levels likely due to more difficulties in coping with the additive effects of our stress model and PHA injection. The lack of IL-1β concentration variations might indicate that sampling immediately post-injection might have been too early to detect any increase, and sampling 24 h post-injection might have been too late [70]. In addition, CO-FES lambs’ plasma IL-2 concentrations were on average smaller than those of ALA-SAIN lambs during the post-IC period. Batuman et al. (1990) [71] found that rats subjected to repeated stress had decreased IL-2 production after PHA stimulation and also decreased mitogen responsiveness of T cells. Coppinger et al. (1991) [72] also found lower IL-2 production in lambs subjected to repeated isolation and restraint. Given the role of IL-2 during the first steps of the cell-mediated immune response, particularly in T cell differentiation [73], results suggest an increased activation of the cell-mediated immunity after PHA stimulation in ALA-SAIN lambs and, in parallel, a lower concentration of cortisol. Considering behavior and immune results, pre-natal supplementation would have resulted in two differentiated response phenotypes to aversive handling, with ALA-SAIN lambs having enhanced behavioral and immune response mechanisms. In addition, plasma IL-10 concentrations post-injection suggest that the downregulation of the pro-inflammatory response to limit tissue damage [74] was similar across pre-natal treatments or, perhaps, had not yet been activated at the time of blood sampling.

Results indicate that the benefits of pre-natal supplementation on the activation of the cell-mediated immune response, in terms of plasma IL-2 concentration, tended to be more apparent in males, whose values tended to be higher 24 h post-injection. Sex differences might be explained by the regulatory role of estrogens on cytokines, reducing IL-2 production and the proliferation of splenic T cells and resulting in higher susceptibility to infection [75]. This would indicate that the benefits of pre-natal PUFA supplementation on the immune response activation after repeated exposure to stress would be limited in females.

## 5. Conclusions

This study highlights the effects of the strategy of supplementation with nutrients rich in essential fatty acids in the final stage of gestation as a management tool that should be explored to promote colostrum quality related to biologically active molecules as CLA and n-3-PUFA for enhancing the immune response of lambs in their first stage of life. In addition, the stressful conditions, and the way in which they occur in terms of intensity and sequence, should be considered in assessing the interaction between prenatal nutrition, immune system stimulation and lambs’ behavior response. Overall, lambs born from mothers supplemented with PUFA, and particularly males, appeared to be better able to tackle the challenges of rearing up to the time immediately after weaning. The reduced response to stress might benefit their adaptive immune system and enhance and regulate the cellular immune response. Although the results do not support the initial hypothesis as a whole, these findings should be considered to update specific management strategies aimed at improving the lambs’ health and welfare in their early stages of life and probably at reducing mortality rates and the use of veterinary medicines, particularly antibiotics.

## Figures and Tables

**Figure 1 animals-12-01780-f001:**
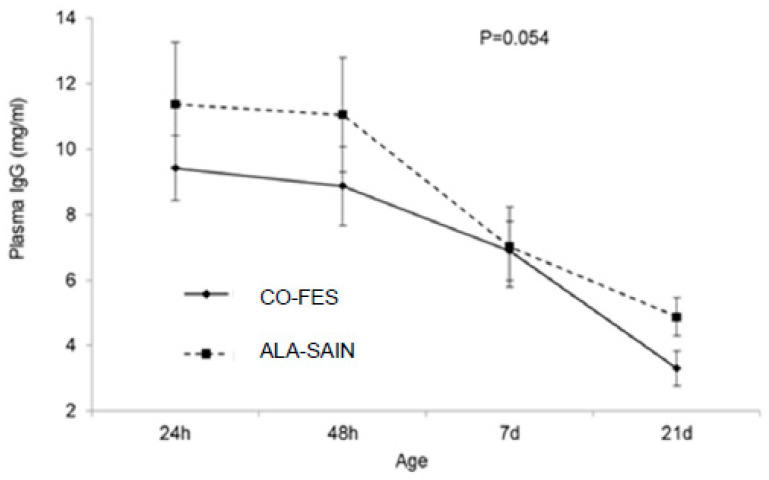
Changes in lamb plasma IgG concentration until 21 d of age according to pre-natal diet.

**Figure 2 animals-12-01780-f002:**
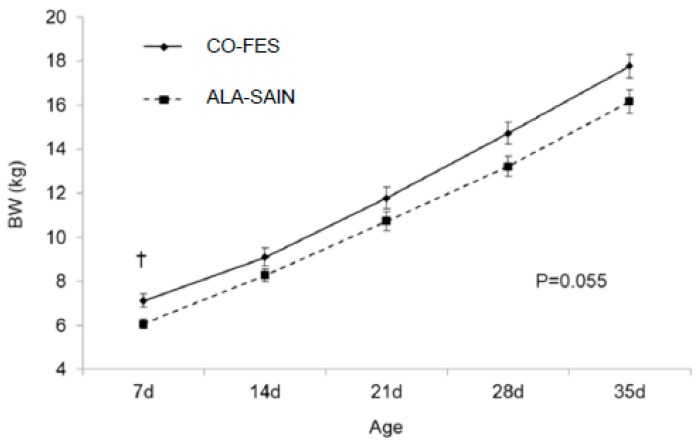
Changes in lamb BW until 35 d of age according to pre-natal diet. † means statistical trend (0.05 < *p* < 0.10).

**Figure 3 animals-12-01780-f003:**
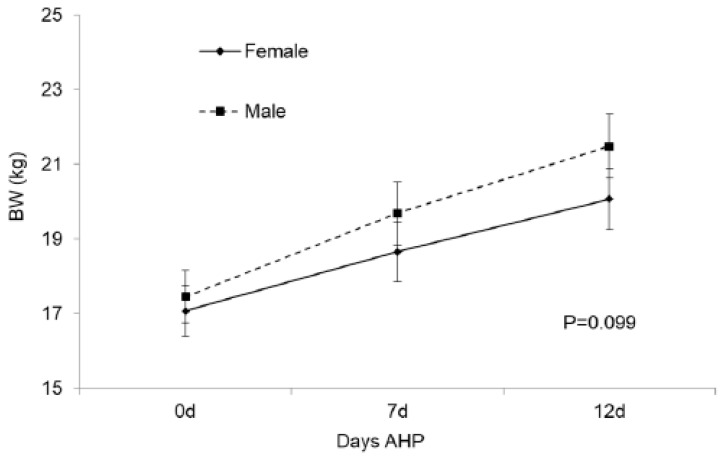
Changes in lamb BW during AHP according to sex.

**Figure 4 animals-12-01780-f004:**
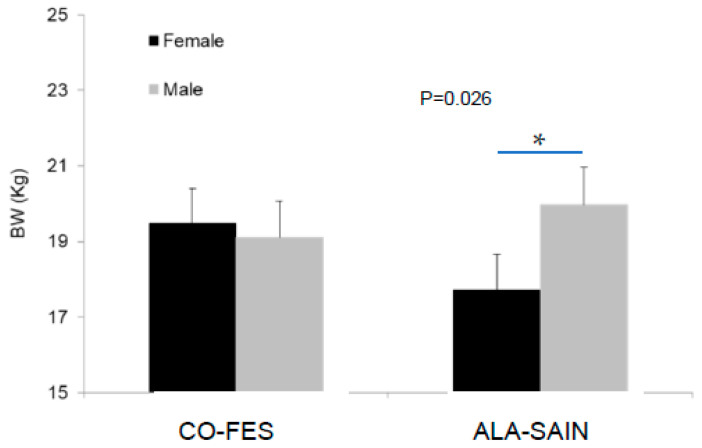
Interaction between pre-natal diet and sex on the mean BW of lambs during AHP. CO-FES—control diet; ALA-SAIN—supplemented diet. Asterisk indicates significant differences between lamb sexes within pre-natal diet.

**Table 1 animals-12-01780-t001:** Effect of pre-natal diet on hematological values of ewes right after birth.

	Pre-Natal Diet				
	CO-FES ^1^		ALA-SAIN ^2^		
Variable	Mean	SE	Mean	SE	*p*-Value
Erythrocytes (×10^6^ cells/mm^3^)	8.56	0.52	9.74	0.57	0.318
Hematocrit (%) ^3^	35.11	1.02	37.25	0.85	0.328
Hemoglobin (g/dl)	10.80	0.54	11.31	0.55	0.578
MCV (fl) ^3,4^	39.82	1.27	38.33	1.00	0.195
Leucocytes (×10^3^ cells/mm^3^) ^3^	5.23	0.77	5.28	0.30	0.887
Eosinophils (%)	0.10	0.37	0.12	0.49	0.979
Neutrophils (%)	47.56	9.24	51.99	8.34	0.803
Lymphocytes (%)	46.37	9.18	41.51	8.12	0.780
Monocytes (%)	5.29	4.05	4.84	3.53	0.953
N:L ^5^	1.46	0.50	1.66	0.45	0.794

^1^ Control diet. ^2^ Supplemented diet. ^3^ Variables for which a lognormal distribution was used. Raw mean (± se) values are presented. ^4^ MCV: mean corpuscular volume. ^5^ Neutrophil lymphocytes ratio.

**Table 2 animals-12-01780-t002:** Effect of pre-natal diet on colostrum physical measurements and chemical composition.

	Pre-Natal Diet				
	CO-FES ^1^		ALA-SAIN ^2^		
Variable	Mean	SE	Mean	SE	*p*-Value
Physical measurements					
Volume (L) ^3^	1410	191	1501	232	0.814
Weight (kg) ^3^	1431	193	1528	235	0.818
Density (g/L) ^3^	1041	1.86	1042	2.19	0.323
pH ^3^	6.47	0.10	6.40	0.08	0.527
Chemical composition					
Fat (%) ^3^	11.99	1.24	11.82	0.97	0.085
Total protein (%)	14.44	0.97	12.22	0.88	0.148
Ash (%) ^3^	0.96	0.04	1.00	0.03	0.628
Lactose (%) ^3^	3.12	0.21	3.33	0.24	0.416
Dry extract (%)	32.26	1.91	27.43	1.74	0.115

^1^ Control diet. ^2^ Supplemented diet. ^3^ Variables for which a lognormal distribution was used. Raw mean (± se) values are presented.

**Table 3 animals-12-01780-t003:** Effect of pre-natal diet on colostrum unsaturated fatty acid profile. Only those unsaturated fatty acids with a contribution of more than 1 mg/g total FA and *p*-value lower than 0.1 are shown.

	Pre-Natal Diet				
	CO-FES ^1^		ALA-SAIN ^2^		
Variable	Mean	SE	Mean	SE	*p*-Value
Unsaturated fatty acids (UFA; mg/g total FA)					
*c*9-14:1 ^3^	1.68	0.26	2.53	0.59	0.303
*t*9-16:1	1.09	0.21	1.83	0.21	0.046
*c*7-16:1	2.15	0.19	1.94	0.20	0.493
*c*9-16:1	9.59	1.28	12.69	1.27	0.150
∑ 16:1 ^3^	13.54	0.61	16.10	1.15	0.042
*c*9-17:1	2.49	0.29	2.23	0.30	0.571
∑ *t6- to t9-*18:1 ^3^	4.61	0.20	4.88	0.33	0.849
*t*10-18:1	4.47	0.30	3.15	0.29	0.014
*t*11-18:1 ^3^	12.47	0.90	25.43	3.14	0.006
*t*12-18:1	2.33	0.19	2.48	0.19	0.615
*t*13- + *t*14-18:1 ^4^	1.10	0.22	2.50	0.22	0.002
*c*9-18:1 ^3^	233.2	13.02	206.1	14.47	0.062
*t*15- + *c*10-18:1	1.54	0.13	1.57	0.13	0.854
*c*11-18:1 ^3^	3.27	0.16	3.78	0.14	0.388
*c*12-18:1	1.45	0.11	2.03	0.13	0.099
*t*16-18:1 ^3^	1.18	0.07	1.78	0.15	0.010
∑ C18:1 *trans*	26.33	3.96	39.71	3.92	0.053
∑ C18:1 *cis* ^3^	237.95	13.22	211.90	14.58	0.069
∑*trans*-MUFA ^3^	27.08	1.50	42.37	4.29	0.041
∑*cis-*MUFA	265.7	16.64	222.3	17.33	0.114
∑MUFA ^3^	284.9	12.9	277.4	13.6	0.235
*c*9, *c*12-18:2 (18:2n-6)	18.04	1.21	19.38	1.19	0.493
*c*9, *c*12, *c*15-18:3 (18:3n-3, ALA)	4.88	0.74	9.82	0.73	<0.001
*c*11, *c*14, *c*17-20:3 (20:3n-3)	2.25	0.12	2.04	0.11	0.260
7c,10*c,13c,16c,19c-22:5* (22:5n-3)	1.74	0.10	1.91	0.10	0.268
4c,7c,10c,13c,16c,19c-22:6 (22:6n-3)	1.27	0.16	1.61	0.16	0.212
∑PUFA	39.50	2.49	52.01	2.46	0.008
*c*9, *t*11-18:2 (*c*9,*t*11-CLA or RA) ^5^	8.56	0.93	13.24	0.92	0.007
∑CLA	9.21	1.01	14.77	1.00	0.005
*Ratio t*11-18:1/*t*10-18:1	3.03	0.49	7.46	0.48	<0.001
∑n-3 ^3^	10.26	0.40	16.10	0.70	0.004
∑n-6	20.10	1.26	21.78	1.25	0.411
n-6:n-3 ^3^	1.91	0.10	1.38	0.05	0.008

^1^ Control diet. ^2^ Supplemented diet. 3 Variables for which a lognormal distribution was used. Raw mean (± se) values are presented. ^4^ coelution with *c*6-/*c*8-18:1. ^5^ coelution with *t*7,*c*9 CLA + *t*8*c*10 CLA. MUFA: monounsaturated fatty acid, PUFA: polyunsaturated fatty acid, CLA; conjugated linoleic acid. ∑*cis*-MUFA: *c*9-10:1; *c*9-12:1; *c*9-14:1, *c*10-15:1, *c*7-16:1, *c*9-16:1, *c*11-16:1; *c*9-17:1, *c*9-18:1, *c*11-18:1, *c*12-18:1, *c*9-20:1, *c*11-20:1, *c*9-24:1; ∑*trans*-18:1: *t*4 + *t*5-18:1; ∑*t*6- to *t*9-18:1, *t*10-18:1, *t*11-18:1, *t*12-18:1, *t*13+*t*14-18:1, *t*15-18:1, *t*16-18:1; ∑*trans*-MUFA: *t*9-16:1 + ∑*t*-18:1; ∑MUFA: ∑*cis*-MUFA + ∑*trans*-MUFA.; ΣCLA: *c*9,*t*11-18:2 coeluted with *t*8,*c*10-18:2 and *t*7,*c*9-18:2; + other isomers (*t,t*-CLA and unknown); ∑n-6: 18:2n-6, 18:3n-6, 20:2n-6, 20:3n-6, 20:4n-6, 22:2n-6; ∑n-3: 18:3n-3, 20:3n-3, 20:5n-3, 22:5n-3 y 22:6n-3; n-3LCPUFA: EPA+DHA; ∑PUFA:∑n-6, ∑n-3, ΣCLA.

**Table 4 animals-12-01780-t004:** Effect of pre-natal diet on colostrum somatic cell counts (SCC), total solids and IgG concentration at birth and 24 h post-lambing.

	Pre-Natal Diet										
	CO-FES ^1^				ALA-SAIN ^2^						
	At Birth		24 h Post-Lambing		At Birth		24 h Post-Lambing				
Variable	Mean	SE	Mean	SE	Mean	SE	Mean	SE	*p*-Value Pre-Natal Diet	*p*-Value Time	*p*-Value Pre-Natal Diet × Time
SCC ^3^ (×10^3^ cells/mL) ^4^	410.7	153.8	198.7	55.7	486.7	204.1	624.5	301.9	0.548	0.587	0.385
Total Solids (° Bx)	23.90	1.25	18.10	1.50	22.03	1.18	14.95	1.18	0.179	<0.001	0.513
IgG (mg/mL)	47.48	3.43	9.65	3.43	40.78	3.41	6.07	3.41	0.214	<0.001	0.627

^1^ Control diet. ^2^ Supplemented diet. ^3^ Somatic cell counts. ^4^ Variables for which a lognormal distribution was used. Raw mean (± se) values are presented.

**Table 5 animals-12-01780-t005:** Effects of pre-natal diet, lamb sex and their interaction on the behavior of lambs during the social isolation and social motivation tests.

	Pre-Natal Diet										
	CO-FES ^1^				ALA-SAIN ^2^						
	Female		Male		Female		Male				
Variable	Mean	SE	Mean	SE	Mean	SE	Mean	SE	*p*-Value Pre-Natal Diet	*p*-Value Sex	*p*-Value Pre-Natal Diet × Sex
Social isolation test											
Passive stand (%) ^4^	58.5	9.3	48.3	9.7	46.5	9.1	41.9	9.5	0.058	0.129	0.612
Move (%) ^4^	17.9	5.0	16.0	4.7	18.0	4.9	18.2	5.4	0.707	0.790	0.736
Explore (%) ^4^	14.8	4.2	16.6	4.6	26.3	5.1	16.7	4.6	0.156	0.385	0.238
Escape attempts (%) ^4^	7.6	4.0	17.8	7.7	7.5	3.7	21.0	0.9	0.738	0.002	0.719
Total distance (m)	2.9	0.5	3.0	0.5	2.6	0.4	2.7	0.5	0.529	0.741	0.974
Angular dispersion ^3^	0.27	0.02	0.33	0.03	0.31	0.04	0.42	0.06	0.170	0.034	0.693
Excretions (n) ^5^	0.0	0.0	0.2	0.2	0.2	0.2	0.3	0.3	0.977	0.977	0.978
Vocalizations (n) ^5^	63.3	6.0	74.3	7.1	73.6	6.7	77.9	7.7	0.096	0.064	0.398
Social motivation test											
Passive stand (%) ^4^	50.4	9.7	51.8	9.9	47.2	9.3	36.6	9.1	0.058	0.325	0.276
Move (%) ^4^	11.3	3.1	11.9	3.3	11.6	2.8	14.4	3.9	0.627	0.576	0.746
Escape attempts (%) ^4^	1.4	2.2	1.3	2.1	0.2	0.4	2.0	3.1	0.288	0.186	0.223
Interact with lambs (%) ^4^	27.1	8.9	21.6	7.8	41.9	10.5	37.9	10.5	0.002	0.266	0.773
Latency to the fence (s) ^3^	2.2	0.4	2.4	0.5	4.2	1.2	1.5	0.3	0.896	0.1337	0.090
Time close to lambs (%) ^4^	92.1 b	5.2	85.3 c	10.2	98.4 a	1.3	89.6 bc	7.6	<0.001	<0.001	<0.001
Total distance (m)	2.0	0.3	1.7	0.3	1.7	0.3	2.4	0.3	0.575	0.451	0.105
Angular dispersion ^3^	0.21	0.01	0.26	0.04	0.33	0.03	0.31	0.07	0.051	0.839	0.251
Excretions (n) ^5^	0.8 a	0.3	0.1 ab	0.3	0.0 b	0.3	0.3 ab	0.3	0.168	0.228	0.022
Vocalizations (n) ^5^	14.8	3.1	11.6	2.7	14.2	3.0	11.0	2.6	0.712	0.079	0.968

^1^ Control diet. ^2^ Supplemented diet. ^3^ Variables for which a lognormal distribution was used. Raw mean (± se) values are presented. ^4^ Variables for which a binomial distribution was used. ^5^ Variables for which a Poisson distribution was used. Within each row, values with different letters indicate significant differences (*p* < 0.05).

**Table 6 animals-12-01780-t006:** Effects of pre-natal diet, lamb sex, time and their two-way interactions on lamb skinfold thickness and skin temperature after the inflammatory challenge post AHP.

	Pre-Natal Diet				Sex				Hours Post-IC ^3^											
	CO-FES ^1^		ALA-SAIN ^2^		Male		Female		24 h		48 h		72 h							
Variable	Mean	SE	Mean	SE	Mean	SE	Mean	SE	Mean	SE	Mean	SE	Mean	SE	*p*-Value Pre-Natal Diet	*p*-Value Sex	*p*-Value Time	*p*-Value Pre-Natal Diet × Sex	*p*-Value Pre-Natal Diet × Time	*p*-Value Sex × Time
Skinfold thickness (mm) ^4^	8.6	0.3	8.7	0.3	9.0	0.3	8.4	0.3	9.7 a	0.4	8.9 a	0.3	7.5 b	0.2	0.523	0.481	<0.001	0.923	0.381	0.626
Skin temperature (°C) ^4^	32.0	0.3	32.7	0.2	32.0	0.2	32.7	0.3	32.6 a	0.3	32.4 a	0.3	32.0 b	0.3	0.823	0.452	0.038	0.171	0.792	0.994

^1^ Control diet. ^2^ Supplemented diet. ^3^ IC: inflammatory challenge. ^4^ Variables for which a lognormal distribution was used. Raw mean (± se) values are presented. Within each line, for each main effect, mean values with different letters indicate statistically significant differences (*p* < 0.05).

**Table 7 animals-12-01780-t007:** Effects of pre-natal diet, lamb sex, time and their 2-way interactions on plasma cortisol and interleukin concentrations after the inflammatory challenge post AHP.

	Pre-Natal Diet				Sex				Hours Post-IC ^3^											
	CO-FES ^1^		ALA-SAIN ^2^		Male		Female		0 h		24 h		48 h							
Variable	Mean	SE	Mean	SE	Mean	SE	Mean	SE	Mean	SE	Mean	SE	Mean	SE	*p*-Value Pre-Natal Diet	*p*-Value Sex	*p*-Value Time	*P*-Value Pre-Natal Diet × Sex	*p*-Value Pre-Natal Diet × Time	*p*-Value Sex × Time
Cortisol (ng/mL) ^4^	83.2	10.1	74.9	8.0	84.5	10.4	74.7	8.0	98.6	11.4	72.8	10.5	65.8	10.5	0.047	0.548	0.051	0.638	0.837	0.766
IL-2 (pg/mL) ^4^	51.4	0.2	53.8	0.9	53.2	1.0	52.1	0.4	52.0	0.6	53.0	1.3	52.7	0.6	0.003	0.084	0.600	0.050	0.306	0.078
IL-10 (pg/mlL) ^4^	14.9	0.5	14.3	0.4	14.1	0.6	15.0	0.4	14.2	0.5	14.3	0.8	15.3	0.6	0.318	0.112	0.300	0.718	0.435	0.454
IL-1β (pg/mL) ^4^	40.1	2.8	43.2	3.9	41.8	3.0	41.5	3.7	47.9	6.2	39.1	1.3	38.0	3.3	0.540	0.682	0.269	0.501	0.789	0.309

^1^ Control diet. ^2^ Supplemented diet. ^3^ IC: inflammatory challenge. ^4^ Variables for which a lognormal distribution was used. Raw mean (± se) values are presented.

## Data Availability

Data generated or analyzed are available from the authors on reasonable request.

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
