# Peer review of "Pre-Partum Supplementation with Polyunsaturated Fatty Acids on Colostrum Characteristics and Lamb Immunity and Behavior after a Mild Post-Weaning Aversive Handling Period"

_animals, 2022, doi:10.3390/ani12141780_

Round 1
Reviewer 1 Report
I will change the Title as follow:
Pre-partum supplementation with polyunsaturated fatty acids on colostrum characteristics and lamb immunity and behaviour
Author Response
We thank the Reviewers for their positive feedback about our manuscript. We have considered each of their comments and modified our manuscript accordingly. This has resulted in an improved version of our manuscript, that we hope it is now acceptable for publication in your Journal. Please, see specific responses below.
Reviewer 1:
Title change
Authors: we partially accept the suggestion to change the tittle. We will keep the title part informing about mild aversive handling. The final title is as follows:
Pre-partum supplementation with polyunsaturated fatty acids on colostrum characteristics and lamb immunity and behaviour after a mild post-weaning aversive handling period
Reviewer 2 Report
The manuscript titled Consequences of pre-partum supplementation with polyunsaturated fatty acids on colostrum characteristics and lamb immunity and behaviour after a mild post-weaning aversive handling period is well written, the statistical n and methods are adequate, conclusions are supported by the data. I have no additional suggestions.
Author Response
We thank the Reviewers for their positive feedback about our manuscript. We have considered each of their comments and modified our manuscript accordingly. This has resulted in an improved version of our manuscript, that we hope it is now acceptable for publication in your Journal. Please, see specific responses below.
Reviewer 2:
The manuscript titled Consequences of pre-partum supplementation with polyunsaturated fatty acids on colostrum characteristics and lamb immunity and behaviour after a mild post-weaning aversive handling period is well written, the statistical n and methods are adequate, conclusions are supported by the data. I have no additional suggestions.
Authors: we thank the Reviewer for the positive comments.
Reviewer 3 Report
General: The research question of whether ewe supplementation pre-partum affects lamb immunity and behavior post-stress is interesting. The manuscript is nicely organized, but some revisions and clarifications would improve the manuscript’s quality. Additionally, too many acronyms are used in this manuscript. Most acronyms (AHP, etc.) are not necessary and make the manuscript difficult to read.
Simple Summary:
Line 27: Insert “the” between “of” and “immune response”.
Abstract:
Line 36: “stressors” should be “stressor”.
Line 50: “lamb’ “ should be “lamb’s”.
Introduction:
No specific comments for the Introduction.
Materials and Methods:
Line 130: Please explain what is meant by “22 out of 47 births could be completely controlled”. Why did the authors remove half of their experimental animals from the study?
Line 131: What is the rationale for discarding the first and last birthed lambs?
Lines 151 – 194: This paragraph is extremely long. Please break this paragraph into smaller paragraphs. It is too difficult to follow in its current state.
Line 226: Why were only three weights taken on the lambs? These three time points are mostly meaningless because weight can vary so much day-to-day.
Lines 228 – 257: Please clarify whether the behavior observations were collected directly or indirectly.
Line 271: Why were IL-2, IL-10, and IL-1B selected for this study? The introduction provides no background on why these interleukins are of particular interest.
Line 273: Please include additional information about the commercial ELISA kit. Did it have a model number? What type of ELISA is used (competitive, sandwich, etc.)?
Results:
Line 306: Should “simple births” be “single births”? If it is meant to read “simple births”, what does that mean?
Line 369: How many lambs were males and how many were females? Since sex was a significant effect or was part of a significant interaction, the reader needs to know how many of each sex were enrolled into the study. The study only had 20 lambs to start with, so if the number of male and female lambs is not balanced, this could impact the results.
Discussion:
Line 645: Replace “levels” with “concentrations”.
Author Response
We thank the Reviewers for their positive feedback about our manuscript. We have considered each of their comments and modified our manuscript accordingly. This has resulted in an improved version of our manuscript, that we hope it is now acceptable for publication in your Journal. Please, see specific responses below.
Reviewer 3:
General: The research question of whether ewe supplementation pre-partum affects lamb immunity and behavior post-stress is interesting. The manuscript is nicely organized, but some revisions and clarifications would improve the manuscript’s quality. Additionally, too many acronyms are used in this manuscript. Most acronyms (AHP, etc.) are not necessary and make the manuscript difficult to read.
Authors: we thank the Reviewer for the positive comments relative to our work. As recommended, we have carefully reviewed the manuscript, and we do not see an excessive use of acronyms. In fact, we use them to shorten the manuscript. Nevertheless, we have deleted some unnecessary AHP acronyms (lines 229, 396, 401).
Simple Summary:
Line 27: Insert “the” between “of” and “immune response”.
Authors: Line 27, word “the” included.
Abstract:
Line 36: “stressors” should be “stressor”.
Authors: Line 37, word “stressor” included.
Line 50: “lamb’ “ should be “lamb’s”.
Authors: Line 51, letter “s” included. Thank you.
Introduction:
No specific comments for the Introduction.
Materials and Methods:
Line 130: Please explain what is meant by “22 out of 47 births could be completely controlled”. Why did the authors remove half of their experimental animals from the study?
Authors: The initial number of ewes supplemented with PUFA in the pre-lambing period was 48, but only 22 lambed in the 6:00 a.m. to 4:00 p.m. time frame. The births that occurred within this time frame could be fully monitored, i.e., blood and colostrum samples could be taken from the dam within one hour after birth and at 24 hours after birth, lambs could be separated and bottle-fed within 18 hours after birth and sampled at 24 hours of birth. There was not enough personnel to cover whole lambing season along 24 hours. This is now explained in the methodology.
Line 131: What is the rationale for discarding the first and last birthed lambs?
Authors: We decided to discard the first and last controlled lambs to equilibrate the groups by sex; 10 males and 10 females. This is now further explained in the methodology.
Lines 151 – 194: This paragraph is extremely long. Please break this paragraph into smaller paragraphs. It is too difficult to follow in its current state.
Authors: We have broken the paragraph into smaller paragraphs. We hope that the text is now easier to follow.
Line 226: Why were only three weights taken on the lambs? These three time points are mostly meaningless because weight can vary so much day-to-day.
Authors: We normally weight lamb weekly. Given that the aversive handing period lasted 12 days, three time points to register lamb’s weights were a good compromise to test BW changes while minimizing lambs’ handling.
Lines 228 – 257: Please clarify whether the behavior observations were collected directly or indirectly.
Authors: the behavior observations were collected directly.
Line 271: Why were IL-2, IL-10, and IL-1B selected for this study? The introduction provides no background on why these interleukins are of particular interest.
Authors: We used PHA, a potent mitogen for the stimulation of cell proliferation and immune system. Therefore, we selected those cytokines mainly related with the regulation of T-cell proliferation and inflammatory functions. We also focused on the selection of cytokines with balanced immune response Th1/Th2. So, we think that the investigation of the lamb’s immune response through the selected cytokines and PHA stimulator would provide interesting information in a context of poor welfare and its relationship with nutrition enriched in PUFA, with effects in inflammatory process. This is now mentioned in the methodology.
Line 273: Please include additional information about the commercial ELISA kit. Did it have a model number? What type of ELISA is used (competitive, sandwich, etc.)?
Authors: Thank you for this comment. We have rewritten the following lines in order to include the additional methodology information: Lines 205-207; lines 275-276.
We think that it is not necessary to include the model numbers or commercial references of the reagents or kits used, because this can extend the text unnecessarily. Nevertheless, we can inform you about the references we used:
Sheep CORTISOL ELISA Kit; CUSABIO ref: CSB-E17045Sh)
Sheep Interleukin 2 (IL-2) ELISA Kit: CUSABIO ref CSB-E11217Sh
Sheep Interleukin 10 (IL-10) ELISA Kit: CUSABIO ref CSB-E12817Sh
Sheep Interleukin 1B (IL-1B) ELISA Kit: CUSABIO ref CSB-E10115Sh
Sheep Leptin ELISA Kit: MyBioSource ref: MBS030760
Sheep Lysozyme ELISA Kit: MyBioSource ref: MBS056991
Results:
Line 306: Should “simple births” be “single births”? If it is meant to read “simple births”, what does that mean?
Authors: Line 310: we have replaced simple by single.
Line 369: How many lambs were males and how many were females? Since sex was a significant effect or was part of a significant interaction, the reader needs to know how many of each sex were enrolled into the study. The study only had 20 lambs to start with, so if the number of male and female lambs is not balanced, this could impact the results.
Authors: In the Material and Methods section, specifically in line 208 (2.2.2. Lambs) we describe the number of lambs that participated in the study, that were balanced according to their sex.
Discussion:
Line 645: Replace “levels” with “values”
Authors: Line 649: “levels” has been replaced by “values”.
Round 2
Reviewer 3 Report
Many thanks to the authors for their revisions. Based on how the authors explained their methodology, only 5 female and 5 male lambs were enrolled per each of the two prenatal treatments. This sample size is far too small to provide any meaningful results. An additional trial is needed.
Author Response
Thank you for your revision and suggestions. As specifically explained in the Materials and Methods section (point 2.2.2. lambs; lines 211-216) the total number of lambs that participated in the experiment were 10 males and 10 females. When designing the experiment, we established a minimum sample size of 10 lambs per prenatal treatment as a realistic balance between statistical power of our experiment and practical constraints of our experimental flock (personnel dedication, lambing monitoring requirements). But it was impossible for us to a-priori determine how many lambs would be born, or their sex. Once we had all lambs, we could have 5 males and 5 females per experimental treatment, achieving a balance of sex within experimental treatment, allowing to overcome statistical constrains coming from unbalance. While we acknowledge that this is not a large sample size, we disagree with the Reviewer in that this is not enough sample size as to draw relevant conclusions. Previous experiments relative to prenatal stress carried out with our experimental flock have had the same problem, but this has not prevented us from drawing relevant conclusions (see for instance Averós et al. 2015, Applied Animal Behaviour Science 163, 98-109). Beyond our research group, other groups working with pre-natal stress in lambs have used similar number of lambs per experimental treatment than in the present study, if not smaller (see for instance Coulon et al. 2011, Physiology & Behavior 103, 575-584). But, again, we acknowledge that the sample size is not large. To consider all of these mentioned aspects, we carried out an statistical analysis that includes the effect of sex on the results, as lamb sex is known to influence behaviour during behavioural tests. In the discussion we have been cautious with our statements, suggesting that prenatal nutrition may produce different effects depending on the sex of the lambs. Although we might agree in that further trials would be necessary to confirm our findings, it is impossible for us to repeat this study.
Xavier Averós, Joanna Marchewka, Ignacia Beltrán de Heredia, Adroaldo Jose Zanella, Roberto Ruiz, Inma Estevez,
Space allowance during gestation and early maternal separation: Effects on the fear response and social motivation of lambs,
Applied Animal Behaviour Science, 2015, 163; 98-109.
https://doi.org/10.1016/j.applanim.2014.11.015.
M. Coulon, S. Hild, A. Schroeer, A.M. Janczak, A.J. Zanella,
Gentle vs. aversive handling of pregnant ewes: II. Physiology and behavior of the lambs,Physiology & Behavior, 2011, 103; 575-584.
https://doi.org/10.1016/j.physbeh.2011.04.010.